# The Role of Microglial Purinergic Receptors in Pain Signaling

**DOI:** 10.3390/molecules27061919

**Published:** 2022-03-16

**Authors:** Hidetoshi Tozaki-Saitoh, Hiroshi Takeda, Kazuhide Inoue

**Affiliations:** 1Department of Pharmacology, School of Pharmacy at Fukuoka, International University of Health and Welfare, 137-1 Enokizu, Okawa 831-8501, Japan; hirotakeda@iuhw.ac.jp; 2Institute for Advanced Study, Kyushu University, 744 Motooka, Nishi-ku, Fukuoka 819-0395, Japan; inoue@phar.kyushu-u.ac.jp

**Keywords:** purinergic signaling, pain, microglia

## Abstract

Pain is an essential modality of sensation in the body. Purinergic signaling plays an important role in nociceptive pain transmission, under both physiological and pathophysiological conditions, and is important for communication between both neuronal and non-neuronal cells. Microglia and astrocytes express a variety of purinergic effectors, and a variety of receptors play critical roles in the pathogenesis of neuropathic pain. In this review, we discuss our current knowledge of purinergic signaling and of the compounds that modulate purinergic transmission, with the aim of highlighting the importance of purinergic pathways as targets for the treatment of persistent pain.

## 1. Introduction

The mammalian nervous system employs various modalities of sensation. One of these, pain, is a necessary experience in life and is produced from the integration of various sensations. Pain is unpleasant in nature; therefore, painful experiences are generally not reinforced and are unlikely to be repeated. Thus, acute pain protects an individual from further injury. The neural substrates of pain perception have been extensively studied, and purinergic pathways have been revealed to have key roles in pain transmission.

The purinergic receptor family is divided into two major families—P1 receptors, which are activated by adenosine, and P2 receptors, which bind purine and pyrimidine nucleotides. P2 receptors are further subdivided into ionotropic P2X receptors and G protein-coupled metabotropic P2Y receptors and are widely distributed in the body, and each subtype has specific roles in the various organs and cell types. Multiple purinergic receptor subtypes are involved in nociceptive circuitry that is initiated from peripherals, through a primary afferent nerve to the spinal cord and brain (Figure 1) [1].

Typically, the first responders in pain transmission are nociceptors in afferent nerves. Specialized sensory nerve fibers are involved in this transmission. Different nerve fibers, which are grouped into Aδ and C fibers, conduct nociceptive stimuli from the peripheral terminals. Pain can be caused by thermal, mechanical, and chemical stimuli, and researchers have discovered effector molecules for each stimulus that trigger or modulate nociceptive signaling [2]. Among these nociceptors, some are sensitive to a specific stimulus, while others are sensitive to multiple types of stimuli; thus, there is an integrated and complex system for nociceptive input [3].

Primary afferent nerves that conduct the nociceptive signals innervate the spinal cord dorsal horn, and peripheral sensory inputs are integrated and processed by the local interneuron network in the spinal cord and upstream brain networks form feedback descending projections into the spinal cord. The central terminals of the nociceptor are somatotopically organized, following ventrodorsal-oriented laminae structures. Most C- and Aδ-nociceptive afferents have synaptic contacts in the superficial laminae (I and II). The signals are relayed to the projection neurons directly and indirectly via complex neural circuitry composed of excitatory and inhibitory interneurons, and the projection neurons convey information to multiple supraspinal sites for pain perception in the brain. The descending modulation by supraspinal structures also occurs in the spinal dorsal horn. It is well documented that the rostral ventromedial medulla (RVM) in the brain stem and the locus coeruleus in the dorsal pons exert both inhibitory and facilitatory effects on spinal dorsal neuronal responses. This bulbospinal projection to the spinal cord integrates inputs from multiple brain regions that are involved in the perception of pain. These brain regions include the mesolimbic reward circuit, and the prefrontal and limbic systems that regulate the affective aspects of pain and emotional and motivational responses. Therefore, the emotional and motivational experience can affect the intensity of pain [4,5,6]. Thus, there are multiple sites of pain modulation—peripheral terminals of nociceptors, central terminals of nociceptors, the spinal circuits that target projection neurons, the brain regions that receive nociceptive input, and the brain regions that project to the spinal cord. Abnormal signaling in these circuits has been implicated in pathological pain, such as chronic pain. The mechanisms modulating neuronal excitability and synaptic efficacy in pain have been extensively studied. Adenosine and purine and pyrimidine nucleotides are biochemical neuro- and gliotransmitters that participate in the modulation of pain. Adenosine can induce both analgesia and hyperalgesia through different subtypes expressed in different parts of the nociceptive pathway. P2X receptors form channels that can contribute to excitatory postsynaptic currents in neurons and trigger other cellular functions in non-neuronal cells. P2Y and adenosine receptors can activate second messenger systems, including calcium, cyclic adenosine monophosphate (cAMP), inositol-1,4,5-trisphosphate, and other signaling molecules through coupled G proteins to modulate cellular activities. We now review our current knowledge of the purinergic receptor contribution in the nociceptive circuit (Figure 1).

## 2. A_1_ Receptors

There are four members of the P1 adenosine receptor family—A_1_, A_2A_, A_2B_, and A_3_ receptors. Adenosine receptors are coupled with different G proteins. Both A_1_ and A_3_ receptors are coupled with Gi proteins, leading to a suppression of adenylate cyclase with a subsequent decrease in cAMP levels. In contrast, A_2A_ and A_2B_ are coupled with Gs proteins, which increase adenylate cyclase activity. Each receptor is also known to activate various intracellular mediators.

Previous studies suggested that systemic adenosine administration has an analgesic effect in preclinical pathological pain models through the A_1_ receptor. However, its broad expression in the body and its varied physiological actions hindered the clinical study of A_1_ receptor-selective ligands for the treatment of pain [7,8,9].

A_1_ receptors modulate synaptic activity presynaptically and postsynaptically to elicit (mainly) inhibitory effects. Presynaptic A_1_ receptors reduce neurotransmitter release via activation of G_i_-proteins that suppress presynaptic calcium currents, while postsynaptic A_1_ receptors do so by causing hyperpolarization through the opening of potassium channels [10]. Immunohistochemically, A_1_ receptors are detected in peripheral sensory neurons and particularly densely in lamina II of the dorsal horn of the spinal cord [11,12]. Antinociception by A_1_ receptor activation is considered to involve the inhibition of synaptic transmission through the elevation of K^+^ conductance in spinal dorsal horn neurons [13] and by peripheral terminal inhibition and presynaptic inhibition at the central terminals of sensory nerve fibers to inhibit nociceptive signal transduction via the release of neurotransmitters in the spinal cord [14]. Activation of A_1_ receptors may have circuit-specific regulatory effects because the selective modulation of excitatory synaptic transmission in the intermediolateral cell column has been observed [15]. Supraspinally, the intra-periaqueductal grey (PAG) injection of 2′-Me-CCPA, a selective A_1_ receptor agonist, reduces pain behavior after intraplantar formalin injection by modulating RVM neuronal activities [16].

Despite the difficulties in developing A_1_ receptor ligands as clinical drugs, researchers have focused on A_1_ receptor partial agonists and positive allosteric modulators, which are expected to reduce adverse effects of full agonists and the excessive perturbation of the intrinsic adenosine system [17]. MIPS521, a positive allosteric modulator, whose binding site on the A_1_ receptor has been structurally determined, was demonstrated to function as an analgesic by stabilizing the complex of the A_1_ receptor with the G-protein [18]. In addition to its antinociceptive effects, significant anxiolytic-like effects of A_1_ receptor positive allosteric modulator TRR469 were reported [19]. The anxiolytic effect of TRR469 is comparable to diazepam but without the sedative effect or locomotor disturbances typical of benzodiazepines. Therefore, A_1_ receptor positive allosteric modulators may have potential in treating the emotional aspect of pain. Structures of A_1_ receptor agonists listed above are shown in Figure 2.

## 3. A_2A_ Receptors

Adenosine A_2A_ receptors are expressed in specific brain regions at pre- and postsynaptic sites on neurons and are also expressed in glia, but there is limited evidence for its expression in spinal cord neurons [20,21]. Peripherally, A_2A_ receptors are detected on immune cells that have anti-inflammatory actions and are, therefore, considered as targets for inflammatory and immune conditions [22].

A_2A_ receptors appear to be expressed on the terminals of primary afferent nerve fibers. Intraplantar injection of CGS21680, an A_2A_ receptor agonist, induces mechanical hyperalgesia, which is reduced in A_2A_ knockout mice and by treatment with the A_2A_ receptor inverse agonist ZM241385 [23]. In another study, SCH58261, a selective A_2A_ antagonist, was shown to suppress nociceptive behavior in animal tests of acute pain, such as the writhing test, the tail-flick test, hot plate test, and the tail immersion test [24,25]. Antinociceptive effects of adenosine A_2A_ receptor antagonists have also been reported when administrated into the cerebral ventricles [26]. The amygdala is one of the key central regions that affect nociception. A_2A_ receptors have been shown to regulate synaptic plasticity in the amygdala, and this property may underlie the antinociceptive role of central A_2A_ receptors. SCH58261 and ZM241385 significantly suppress high frequency stimulation-induced long-term potentiation in the amygdala, and this action may contribute to the ability of the A_2A_ receptor antagonists to attenuate contextual fear memory [27].

The accumulating evidence suggests a pronociceptive effect of A_2A_ receptors in peripheral nerves. However, in non-neuronal cells, the anti-inflammatory effect of A_2A_ receptor agonists seems to have antinociceptive effects in some types of chronic pain. Long-term antinociceptive effects of intrathecal A_2A_ agonists, such as ATL313 and CGS21680, have been demonstrated in a model of neuropathic pain that involves massive activation of microglia and astrocytes in the spinal cord after peripheral nerve injury [28]. Nerve injury induces both the activation and proliferation of microglia and astrocytes responsible for inflammation in neuropathic pain. The upregulation of glial activation markers is persistently reduced after a single treatment of A_2A_ receptor agonist, and the upregulation of tumor necrosis factor-α (TNF-α), a pro-inflammatory cytokine, is suppressed, while the anti-inflammatory cytokine interleukin (IL)-10 remains upregulated in the spinal cord [29]. Therefore, the current evidence supports dual, opposing roles of A_2A_ receptors—as pronociceptive modulators and anti-inflammatory regulators—in the central nervous system (CNS) glial cells. Some allosteric modulators of A_2A_ receptors have been identified and studied concerning inflammatory regulation [30,31]. Future studies establishing the physiologic functions of A_2A_ receptors are awaited. Structures of A_2_ receptor agonists listed above are shown in Figure 2.

## 4. A_3_ Receptors

A_3_ receptors are highly expressed in immune cells, including glial cells [32]. A_3_ receptors in microglia participate in chemotaxis toward ATP [33]. Selective A_3_ receptor agonists, such as IB-MECA and Cl-IB-MECA, are receiving attention as candidate drugs for treating inflammatory diseases, such as rheumatoid arthritis [34]. Interestingly, reciprocal regulation of A2AR and A3R carries an inhibitory effect of adenosine against innate immune response in both activated and homeostatic microglia [35]. A_3_ receptor agonists have also been studied in neuropathic pain [36]. Intraperitoneal injections of IB-MECA suppressed neuropathic pain states by reducing spinal microglial activation [37]. IB-MECA, Cl-IB-MECA, and MRS1898 have been shown to alleviate neuropathic pain induced by chronic constriction injury and chemotherapeutics [38]. A_3_ receptor activation by IB-MECA suppresses nicotinamide adenine dinucleotide phosphate (NADPH) oxidase activity, the production of pro-inflammatory cytokines (TNF-α, IL-1β), and upregulates the anti-inflammatory cytokine IL-10 [39]. MRS5698, another agonist, reverses persistent neuropathic pain without tolerance, as is observed with continuous morphine treatment. Interestingly, bilateral RVM microinjections of MRS5698 have an analgesic effect on chronic constriction injury (CCI)-induced neuropathic pain. Thus, supraspinal A_3_ receptors can also contribute to the amelioration of neuropathic pain [40]. A recent study demonstrated that the antiallodynic effect of MRS5980 depends on CD4^+^ T cells, which interact with dorsal root ganglion cells in an IL-10-dependent manner [41]. Another study reported that MRS5980 prevents cisplatin-induced cognitive impairments, sensorimotor deficits, and neuropathic pain [42]. A_3_ receptor agonists (Figure 2) already demonstrate safety profiles in clinical trials for cancer treatment. Clinical application of A_3_ receptor agonists can be expected.

## 5. P2X2 and P2X3 Receptors

P2X2 and P2X3 are expressed by small sensory neurons of the dorsal root ganglia (DRG), which has been confirmed by ATP-induced currents in DRG neurons [43]. P2X2 and P2X3 homomeric channels can be discriminated by their rapid and slow desensitization to ATP stimulation, respectively, and by the insensitivity of P2X2 to α,β-methylene ATP [44,45]. P2X2 and P2X3 can form heteromeric P2X2/3 channels, which have mixed current properties [46,47]. Expression of these receptors has been demonstrated in rodent DRGs, but primate DRGs appear to only express P2X3 [48]. A recent study using P2X2 reporter mice showed that P2X2 expression is lower in DRGs and trigeminal ganglia than the level suggested by past studies [49].

P2X3 knockout mice have normal noxious mechanosensation and acute pain responses, but extracellular recordings in the dorsal horn indicate loss of electrical responses to a temperature change of around 40 °C [50]. As formalin-induced pain behavior is significantly reduced in P2X3 knockout mice, a number of studies have examined the potential of P2X3 receptor antagonists for the treatment of chronic pain [50,51]. Various mechanisms of P2X3 receptor modulation have been reported in the hyperalgesic state in chronic pain [52].

A-317491 is a non-nucleotide antagonist that blocks P2X3 and P2X2/3 channels. A-317491 suppresses mechanical allodynia in a model of neuropathic pain and reduces formalin-induced pain in knockout mice [53]. Gefapixant (MK-7264), an orally bioavailable P2X3 and P2X2/3 receptor antagonist, which has been approved for the treatment of refractory or unexplained chronic cough, was also demonstrated to relieve inflammatory, osteoarthritic and neuropathic pain [54]. Eliapixant (BAY-181780), another potent and relatively selective P2X3 homomer antagonist undergoing clinical trial, also reduces inflammatory pain in preclinical models [55]. Two other non-competitive P2X3 homotrimeric receptor antagonists, BLU-5937 and sivopixant (Figure 3), have a higher selectivity for the P2X3 versus P2X2/3 and are under clinical trial for the treatment of refractory chronic cough [56,57]. Sivopixant showed a strong analgesic effect in the rat partial sciatic nerve ligation model [58]. A benzimidazole-4,7-dione analog, KCB-77033, was newly identified and showed pain relief in a cisplatin-induced neuropathic pain model [59].

## 6. P2X4 Receptors

The central role of the P2X4 receptor in pain was first reported by Tsuda et al. [60], who demonstrated that intrathecal P2X4 receptor antagonist injection alleviated allodynia in a rat model of neuropathic pain. This study suggested that the activation of the P2X4 receptor on spinal microglia is both necessary and sufficient to induce tactile allodynia after peripheral nerve injury. The activated spinal microglia strongly expressed P2X4 receptors in their neuropathic pain model. Furthermore, intraspinal transplantation of microglia following P2X4 receptor stimulation induced hyperalgesia. Together, these findings demonstrate a critical role of spinal microglia in the modulation of pain.

The current consensus is that microglial surface P2X4 receptors are upregulated following inflammatory activation. Two independent research groups generated reporter mice showing P2X4 receptor expression in the brain, and both reported similar observations. Under physiological conditions, P2X4 receptors are expressed in various neurons in the brain but not in microglia, and expression in microglia is seen only after lipopolysaccharide (LPS) treatment [61,62]. Notably, in one of the studies, a small group of reporter gene-positive neurons in the hypothalamus did not show an electrophysiological response to ATP. This suggests the absence of functional surface P2X4 receptors in neurons, despite the presence of transcriptional activity. The other study showed the upregulation of surface P2X4 receptors on microglial cells by inflammatory stimuli through lysosomal secretion [63,64]. These findings suggest the possibility of targeting pathogenic P2X4 receptors in activated microglia.

If we consider microglial P2X4 upregulation as a form of microglial activation, many factors can regulate pain-related P2X4-positive microglial activation. Interferon regulatory factor (IRF) 5 is a direct transcriptional regulator of P2X4 receptors and is in turn under IRF8 transcriptional control [65]. MafB, whose expression is regulated by miro RNA, mir-152-3p, -mediated translational regulation is also involved in P2X4 receptor upregulation [66]. These pain-related microglial P2X4 receptors are known to induce brain-derived neurotrophic factor (BDNF) production and release from microglia to modulate pain-transducing spinal circuits [67,68].

Developed as a negative allosteric modulator of P2X4 receptors, 5-BDBD, a benzodiazepine derivative, has been demonstrated to reduce pain behavior caused by chronic constriction injury in animal studies [69,70]. NP-1815-PX, which is more water-soluble and has high potency for human P2X4 receptors, was shown to inhibit mechanical allodynia in HSV-1-inoculated mice without affecting normal pain sensitivity or motor function [71]. NC-2600 has completed phase I clinical trials for neuropathic pain and is awaiting further study [72]. PSB-15417 is a potent, blood–brain barrier (BBB)-permeable, allosteric modulator of P2X4 receptors that has an analgesic effect in animal models of neuropathic pain [73]. BAY-1797 is an orally active and selective P2X4 antagonist with antinociceptive and anti-inflammatory effects in the mouse complete Freund’s adjuvant (CFA) inflammatory pain model [74]. Interestingly, some antidepressants antagonize recombinant human and rat P2X4 receptors [75], and duloxetine, which is recommended for some types of chronic pain, might alleviate chronic pain through P2X4 receptor inhibition [76]. These findings suggest that the P2X4 receptor may be a potential target for psychiatric therapy. Structures of P2X4 receptor antagonists listed above are shown in Figure 3.

Neuronal P2X4 receptors can modulate activity-dependent plasticity at central synapses, such as long-term potentiation (LTP) and long-term depression (LTD), without altering basal activity [61,77]. The ethanol sensitivity of P2X4 receptors (reviewed in [78]) suggests that alcohol may impact brain function in part by affecting these receptors. P2X4-deficient pups, which have altered hippocampal glutamate receptor composition, show significant reductions in social interaction and ultrasonic vocalizations [79]. Mice in which the P2X4 receptor is genetically replaced with a recombinant P2X4 receptor that shows elevated surface expression on excitatory neurons exhibit increased time in the open arm in the elevated plus maze and increased time in the center zone of the open-field test. This anxiolytic phenotype is abolished when the recombinant receptors are introduced into native P2X4-expressing cells (microglia) by Cre-mediated recombination driven by a ubiquitous promoter, CMV [61]. Therefore, the P2X4 receptor has a cell-type-specific role in the emotional activity, and depression as a comorbid disorder after chronic pain could be associated with the P2X4 receptor.

## 7. P2X7 Receptors

It has been proposed that the P2X7 receptor is absent in neurons but expressed in glial cells [80]. The recent development of genetic modulation techniques has clearly revealed the pattern of P2X7 receptor expression in the CNS. Bacterial artificial chromosome (BAC) transgenic mice harboring an enhanced green fluorescent protein (EGFP)-tagged P2X7 receptor gene instead of the native *P2rx7* gene demonstrate dominant P2X7-EGFP protein expression in microglia, satellite glia ensheathing DRG neurons, oligodendrocytes, and a limited population of astrocytes but not in neurons [81,82]. In addition, the reliability of a newly developed highly specific anti-P2X7 receptor nanobody and inadequate binding of commercially available anti-P2X7 antibodies has been shown [83].

As mentioned before, microglia critically contribute to neuropathic pain. P2X7 receptors in microglia might be involved in the production and release of cytokines and chemokines, such as IL-1β, IL-6, TNF-α, chemokine (C-C motif) ligand 3 (CCL3), and chemokines (C-X-C motif) ligand 2 (CXCL2) [84,85,86,87,88]. Thus, the activation of microglial P2X7 triggers the formation of a proinflammatory environment in the CNS. Antagonists selective for P2X7 receptors are accordingly expected to suppress neuroinflammatory processes.

P2X7 receptor knockout mice exhibit reduced pain hypersensitivity in a model of neuropathic pain [89]. Blockade of P2X7 receptors significantly reduces nociception in animal models of chronic neuropathic and inflammatory pain [90,91]. Rodent models of chronic inflammatory pain show alleviation of pain by P2X7 receptor antagonists, such as oxidized ATP, A-740003, and A-438079 [92,93,94]. A438079 also suppresses paclitaxel-induced mechanical hypersensitivity, which can also be alleviated by a CCL3-neutralizing antibody [95]. It was reported that intra-amygdala infusion of A-438079 in a neuropathic pain model reduces depression- and anxiety-like behaviors, along with an antinociceptive effect [96]. JNJ-47965567, a CNS-permeable P2X7 antagonist, has the ability to prevent mechanical hypersensitivity in a rat model of neuropathic pain [97]. Notably, CNS-permeable P2X7 antagonists, such as JNJ-55308942 and JNJ-54175446, were chosen as clinical candidates for major depression because of their anti-neuroinflammatory effects [98,99]. Structures of P2X7 receptor antagonists listed above are shown in Figure 3.

Restraint stress increases ATP, IL-1β, and TNFα in the hippocampus. Furthermore, A-804598, a P2X7 receptor antagonist, blocks the induction of IL-1β and TNFα [100]. ATP and glutamate release in hippocampal slices from stressed mice is mediated by Cx43 and Panx1 hemichannel activation via N-methyl-D-aspartic acid/P2X7 receptor signaling. Chronic, but not acute, restraint stress upregulates Panx1 in astrocytes and neurons in the hippocampus [101,102]. Whether pain experience evokes gliotransmitter release in the brain is still unknown; however, targeting the P2X7 receptor to treat chronic pain-associated depression may be worth investigating in future studies.

## 8. P2Y_1_ Receptors

P2Y receptors are G-protein-coupled receptors that are stimulated by purine and pyrimidine nucleotides. The P2Y_1_ receptor is widely distributed in the CNS. In the nociceptive circuitry, P2Y_1_ receptors can be detected in small-diameter sensory neurons in the DRG, in a subset of transient receptor potential V1 (TRPV1) receptor-positive cells [103,104]. It has been suggested that P2Y_1_ and/or P2Y_2_ receptor activation modulates TRPV1 responses by lowering the activation threshold for capsaicin, protons, and heat stimulation, contributing to ATP-induced hypersensitivity [105]. In the RVM, ATP activates off-cells, whose activation is associated with antinociception. This activation is antagonized by the selective P2Y_1_ receptor antagonist MRS2179 [106]. In the spinal cord, expression of the P2Y_1_ receptor can be detected [107,108]. MRS2179, a P2Y_1_ receptor-specific inhibitor, was demonstrated to reverse mechanical hypersensitivity and spontaneous pain in the rodent model of cancer-induced bone pain when intrathecally administered [109]. MRS2365 and MRS2500, another P2Y_1_ receptor-selective antagonists (Figure 4), were shown to have an analgesic effect on inflammatory pain [110,111,112].

## 9. P2Y_2_ Receptors

P2Y_2_ is expressed in DRG and trigeminal neurons. Activation of the P2Y_2_ receptor by uridine triphosphate (UTP) triggers action potential firing in dissociated DRG neurons and induces cAMP response element-binding protein (CREB) phosphorylation [113]. UTP activates the terminals of the majority of C fibers and a small population of Aβ fibers, which are also activated by capsaicin in mouse skin nerve preparations [114]. In another report, P2Y_2_ receptor expression, by immunolabeling, was observed in satellite glia of the trigeminal ganglion, and AR-C118925 (Figure 4), a P2Y_2_ receptor-selective antagonist, reversed facial allodynia in the complete Freund’s adjuvant (CFA)-induced chronic pain model [115]. In the ophthalmic field, a P2Y_2_ receptor agonist, diquafosol, has been approved to treat dry eye disease.

## 10. P2Y_12_ Receptors

P2Y_12_ receptors are coupled with Gi proteins and respond to adenosine diphosphate (ADP), and play key roles in platelet activation. Four specific antagonists of P2Y_12_ receptors have been approved as antithrombotic agents—clopidogrel, prasugrel, cangrelor, and ticagrelor [116]. Selatogrel, a novel reversible P2Y_12_ receptor antagonist, is under clinical trial [117]. The P2Y12 receptor in the CNS is a marker of homeostatic microglia and is considered to be a critical receptor in the surveillance of the local incidents in the CNS by regulating the motility of ramified processes [118].

It was demonstrated that intrathecally administered P2Y_12_ receptor antagonists, such as MRS2395, AR-C69931MX, and PSB-0739 (Figure 4), significantly suppress mechanical hypersensitization in models of neuropathic pain and inflammatory pain [119,120,121]. P2Y12 knockout mice have provided valuable insight into the role of these receptors. Genetic ablation of P2Y12 receptors mitigates inflammatory and neuropathic pain [119,122,123].

Microglial P2Y_12_ expression is reduced in neuroinflammatory CNS diseases, such as multiple sclerosis and Alzheimer’s disease [124,125,126]. However, some reports indicate spinal P2Y_12_ receptor upregulation in models of chronic pain [127]. Therefore, reactive spinal microglia in models of chronic pain might undergo a specific mode of activation in the development of hypersensitivity. Several studies have shown that the P2Y_12_ receptor activates p38 MAPK, via RhoA/ Rho-associated coiled-coil containing protein kinase 2 (ROCK2) signaling [128,129]. Both ROCK2 inhibitor and P2Y_12_ receptor antagonists suppress p38 phosphorylation and show analgesic effects on the neuropathic pain model. MRS2395 also reduces the GTP-bound form of RhoA and suppresses ROCK2 upregulation in the spinal cord. Signaling under the P2Y_12_ receptor includes RhoA/ROCK2-mediated p38 phosphorylation to induce neuropathic pain. It must be noted that P2Y_12_ receptors are predominantly expressed in platelets in the periphery and that antagonists have antithrombotic effects. Microglia-specific analysis of P2Y_12_ receptor signaling is important for the further development of P2Y_12_-related analgesic agents.

## 11. Other Purinergic Molecules

Extracellular adenosine can be produced extracellularly as a metabolite of the ATP released by cells under stressful conditions. Dephosphorylation by two hydrolyzing enzymes, CD39 and CD73, is primarily involved in this process [130,131]. Prostatic acid phosphatase (PAP) and tissue-nonspecific alkaline phosphatase (TNAP) are also known to hydrolyze extracellular adenosine monophosphate (AMP) to adenosine (Figure 5) [132]. As adenosine is involved in nociceptive regulation, the effect of modulating adenosine metabolism on pain has been examined. Intrathecal administration of recombinant enzymes, such as CD73 and PAP, revealed an increase in adenosine concentration in the spinal cord subsequent analgesic effect in inflammatory pain that was lost with A_1_ receptor deficiency [132,133,134]. Therefore, A_1_ receptor-mediated inhibitory effect on the spinal nociceptive circuit (shown in Section 2) can be enhanced by exogenous ectonucleotidases introduction. Other studies demonstrate that adenosine kinase (AK) inhibition increases extracellular concentrations of adenosine, leading to adenosine receptor-mediated analgesia in pathological pain [135,136]. Microglia may be important players in the CD39–CD73–adenosine receptor signaling axis because microglia express functional levels of CD39 and CD73 in vivo and have major roles in regulating neuronal activity in the brain [137,138].

Vesicular nucleotide transporter (VNUT; also known as Slc17a9), which is involved in ATP storage and release, is an important regulator of pathological sensory hypersensitivity. VNUT knockout mice exhibit suppressed mechanical hyperalgesia, and cell-type-selective gene knockdown using the Cre–loxP recombination system showed that neuronal VNUT in the spinal dorsal horn is responsible for mechanical hypersensitivity in a neuropathic pain model [139]. Clodronate is identified as a potent and selective VNUT inhibitor and has been shown to suppress inflammatory pain and neuropathic pain [140].

ATP can be released into the extracellular space through vesicular exocytosis at the synapse. A vesicular nucleotide transporter (VNUT) is critical transporter for ATP storage. Endogenous ecto-ATPases on cell membrane hydrolyze ATP to adenosine diphosphate (ADP) and adenosine monophosphate (AMP), and the hydrolysis of AMP by CD73, tissue-non-specific alkaline phosphatase (TNAP), and prostatic acid phosphatase (PAP) produce extracellular adenosine.

## 12. Conclusions and Prospects

In this paper, we reviewed purinergic receptors ligands, their effects on the nociceptive system, and their sites of action in the nociceptive pathway (Figure 1). Purinergic ligands, purine and pyrimidine nucleotides, and adenosine have very unique and elaborately created metabolic systems in the body. The receptor family is widely distributed in the body, and each subtype has specific roles in the various organs and cell types. At peripheral sites, neuronal P2X3 receptors may contribute to the initiation of acute nociception and acute inflammatory pain. P2Y_1_ and P2Y_2_ receptor activation in DRGs can modulate the activity of TRPV1^+^ nociceptive neurons. Several purinergic receptors, such as P2X7 and P2Y_2_ receptors, are expressed in satellite glial cells surrounding the cell bodies of sensory neurons that modulate the activity of nociceptors. Adenosine, an ATP metabolite, can induce hyperalgesia through peripheral A_2A_ receptor activation. In the spinal cord, P2X4, P2X7 and P2Y_12_ receptors on microglia contribute to the development of neuropathic pain through complex neuronal–glial interactions, and anti-inflammatory effect of A_2A_ and A_3_ receptors may counteract the painful activation of spinal microglia. A_1_ receptor activation can block excitatory nociceptive transmission in the dorsal spinal cord circuit. Supraspinally, A_3_ receptors in the RVM may contribute to mechanical hypersensitivity in neuropathic pain. Multiple purinergic receptors can affect the activity of the nociceptive neural network at multiple sites, and the effect manifested may differ in physiological and pathophysiological conditions.

We may be far from a complete understanding of this complex system in nociceptive circuitry, but there has steady accumulation of scientific knowledge and materialization. Purinergic signaling is an important therapeutic target for the treatment of pathological pain. Novel agonists, antagonists, and allosteric modulators of purinergic signaling only await discovery. For effective analgesic agents, however, a systematic and deeper understanding of the role of purinergic signaling in the sensory and emotional aspects of pain is required.

## Figures and Tables

**Figure 1 molecules-27-01919-f001:**
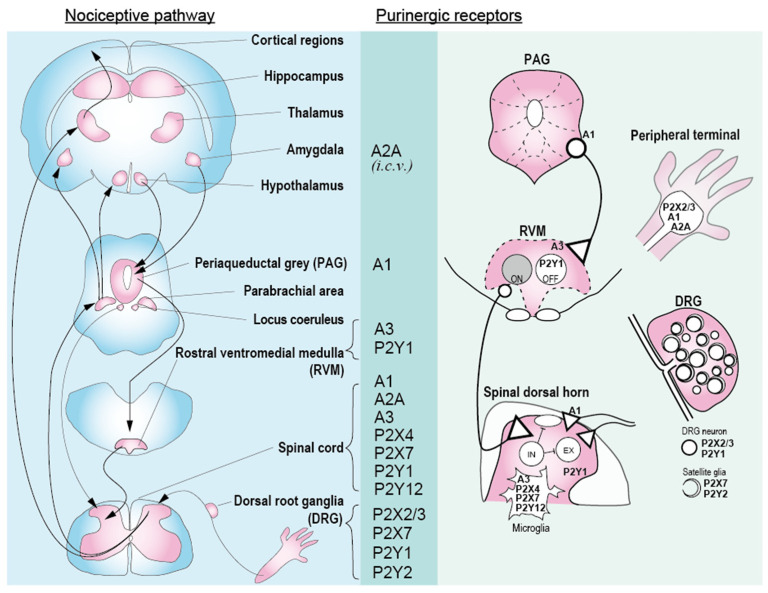
Major nociceptive pathway from peripheral to central nervous system (rodent’s structure). Main ascending (peripheral–dorsal root ganglia (DRG)–spinal cord–parabrachial area–thalamus and amygdala) and descending (hypothalamus and amygdala–periaqueductal grey (PAG)–rostral ventromedial medulla (RVM)–spinal cord) nociceptive pathways are shown on the left. The key regions for pain modulation are labeled. Purinergic receptors demonstrated to modulate nociceptive responses are listed in the center. Right indicates detailed expression pattern of the receptor, as mentioned in this review.

**Figure 2 molecules-27-01919-f002:**
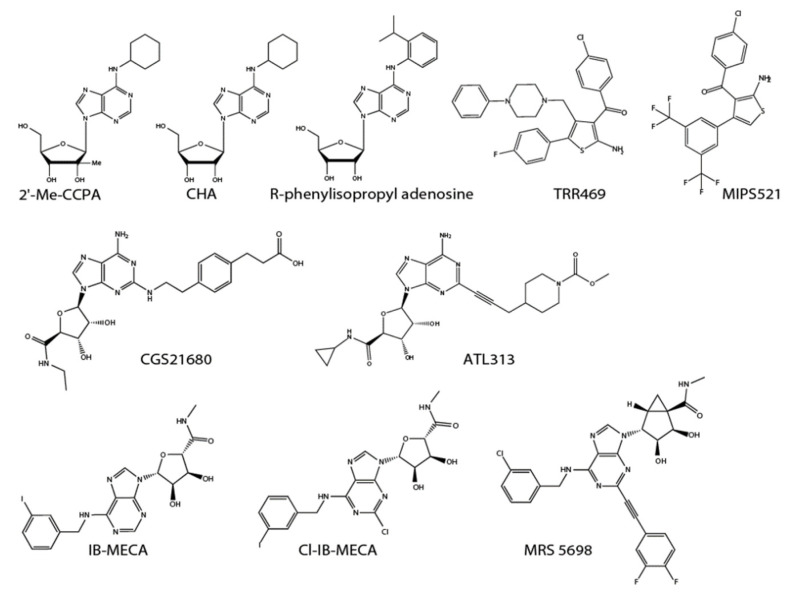
Structures of listed P1 receptors agonists and allosteric modulators.

**Figure 3 molecules-27-01919-f003:**
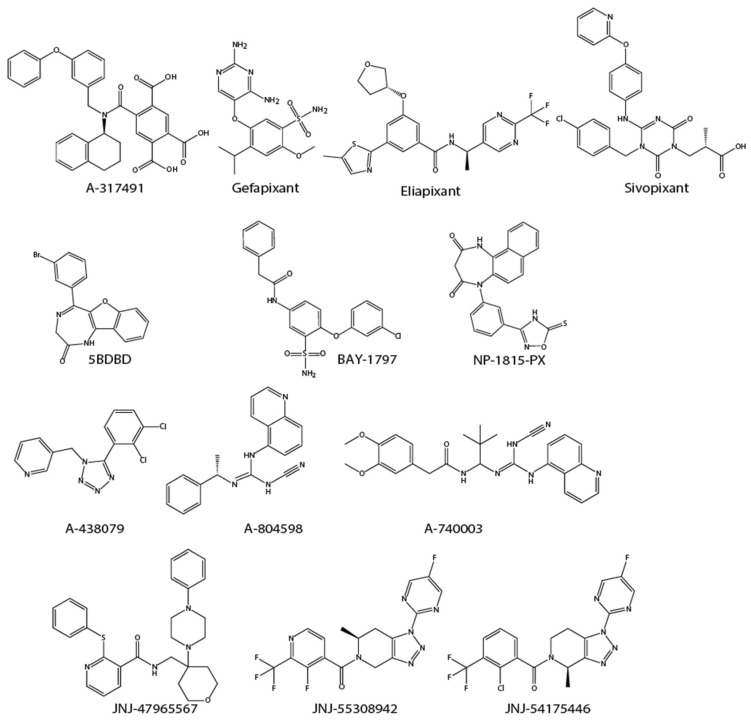
Structures of listed P2X receptors antagonists. Chemical structures of PSB-15417 and NC2600 have not been disclosed.

**Figure 4 molecules-27-01919-f004:**
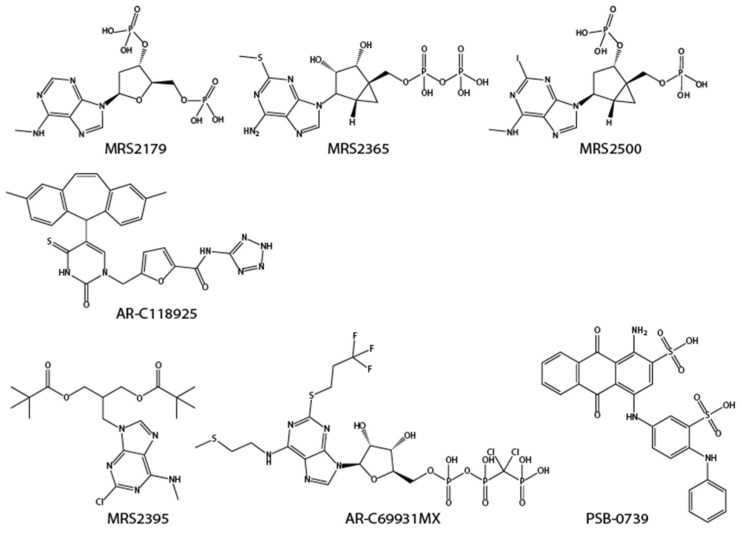
Structures of listed P2Y receptors antagonists.

**Figure 5 molecules-27-01919-f005:**
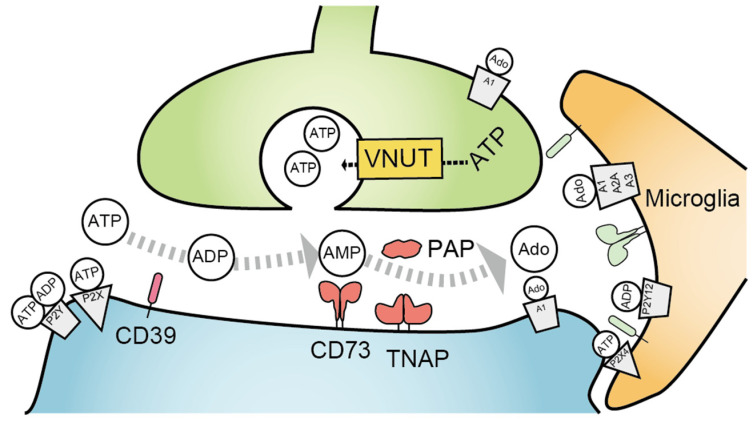
The process of adenosine triphosphate (ATP) release and metabolism at extracellular space.

## Data Availability

Not applicable.

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
