# Peer review of "The Role of Microglial Purinergic Receptors in Pain Signaling"

_molecules, 2022, doi:10.3390/molecules27061919_

Round 1

Reviewer 1 Report

Manuscript entitled „The role of microglial P2 receptors in pain signaling” is an interesting, well-written and well-planned review work. However, the text needs some corrections according to the following comments:

  1. Introduction

line 60 – explain in full name the abbreviation cAMP

  1. A2A receptors

line 130 - explain in full name the abbreviation TNF – α

line 131 - explain in full name the abbreviation IL

line 133 - explain in full name the abbreviation CNS

  1. A3 receptors

line 141 - explain in full name the abbreviation NADPH

  1. P2X2 and P2X3 receptors

line 152 - explain in full name the abbreviation DRG

line 165 – should be [42, 43]

  1. P2X4 receptors

line 191 - explain in full name the abbreviation LPS

line 202 – explain what MafB and mir-152-3p means? What is it?

line 204 - explain in full name the abbreviation BDNF

line 212 - explain in full name the abbreviation BBB

  1. P2X7 receptors

line 247 - explain what CCL3 and CXCL2 means? What is it?

line 264 – live only abbreviation ATP, it should be written in full name earlier in the text

line 267 - explain in full name the abbreviation NMDA

line 277 - explain in full name the abbreviation TRPV1

  1. P2Y2 receptors

line 291 - explain in full name the abbreviation UTP and CREB

line 296 - explain in full name the abbreviation CFA

  1. P2Y12 receptors

line 301 - explain in full name the abbreviation ADP

line 314 - give examples of literature confirming this sentence:

“Microglial P2Y12 expression is reduced in neuroinflammatory CNS diseases, such as 313 multiple sclerosis and Alzheimer’s disease”.

line 318 – explain this “p38 MAPK, via GTP-RhoA/ROCK2 signaling”

  1. Other purinergic molecules

line 329 - explain in full name the abbreviation AMP

line 332 – explain the function of receptor A1

  1. Conclusion and prospects

line 355 and 357- there is nothing in the main text about the function of A2a and A3 receptors. Please include them and add relevant literature citations

Figure 1 and Figure 2- all abbreviations used in the description of the figure or on the figure should be explained with the full name in accordance with the journal's requirements:

Acronyms/Abbreviations/Initialisms should be defined the first time they appear in each of three sections: the abstract; the main text; the first figure or table. When defined for the first time, the acronym/abbreviation/initialism should be added in parentheses after the written-out form.

Additionally, the legend for the figure 1 and figure 2 should be placed directly below it

Author Response

Response to Reviewer 1

Comments and Suggestions for Authors

Manuscript entitled „The role of microglial P2 receptors in pain signaling” is an interesting, well-written and well-planned review work. However, the text needs some corrections according to the following comments:

We are grateful to Reviewer 1 for reviewing our manuscript and giving worthful comment. Below are our responses to each comment. We believe improvement of our manuscript by your kind suggestions.

Introduction

line 60 – explain in full name the abbreviation cAMP*

A2A receptors

line 130 - explain in full name the abbreviation TNF – α*

line 131 - explain in full name the abbreviation IL*

line 133 - explain in full name the abbreviation CNS*

A3 receptors

line 141 - explain in full name the abbreviation NADPH*

P2X2 and P2X3 receptors

line 152 - explain in full name the abbreviation DRG*

line 165 – should be [42, 43]*

P2X4 receptors

line 191 - explain in full name the abbreviation LPS*

line 202 – explain what MafB and mir-152-3p means? What is it? *

line 204 - explain in full name the abbreviation BDNF*

line 212 - explain in full name the abbreviation BBB*

P2X7 receptors

line 247 - explain what CCL3 and CXCL2 means? What is it? *

line 264 – live only abbreviation ATP, it should be written in full name earlier in the text*

line 267 - explain in full name the abbreviation NMDA*

line 277 - explain in full name the abbreviation TRPV1*

P2Y2 receptors

line 291 - explain in full name the abbreviation UTP and CREB*

line 296 - explain in full name the abbreviation CFA*

P2Y12 receptors

line 301 - explain in full name the abbreviation ADP*

line 314 - give examples of literature confirming this sentence: *

“Microglial P2Y12 expression is reduced in neuroinflammatory CNS diseases, such as 313 multiple sclerosis and Alzheimer’s disease”.

line 318 – explain this “p38 MAPK, via GTP-RhoA/ROCK2 signaling” *

Other purinergic molecules

line 329 - explain in full name the abbreviation AMP*

line 332 – explain the function of receptor A1*

We appreciate and agree with the reviewer’s point. We have added the full names of abbreviations and additional explanation of the sentences pointed out.

Conclusion and prospects

line 355 and 357- there is nothing in the main text about the function of A2a and A3 receptors. Please include them and add relevant literature citations

We appreciate and agree with the reviewer’s point. We have added a description of the anti-inflammatory function of microglial A2a and A3 in line 160-162 of the revised text.

Figure 1 and Figure 2- all abbreviations used in the description of the figure or on the figure should be explained with the full name in accordance with the journal's requirements:

We appreciate and agree with the reviewer’s point. We have added the full names of abbreviations in the figure legend.

Reviewer 2 Report

This review article addresses the role of microglial P2 receptors in pain signaling. This article discussed the importance of purinergic signaling. It also reviewed some novel purinergic agents that might control chronic pain. In my opinion, this manuscript is rigorous and provides a useful contribution to its area of research. I only have some minor queries that require additional attention.  

  1. Title: I considered that the role of microglial P2 purinergic receptors in pain signaling could be better.
  2. Introduction: Please add brief sentences to describe the classification of the P1 and P2 receptors. Additionally, the relationship between P1 (not just P2) purinergic receptors and pain sensation is also required in this section.
  3. The abbreviation of RVM in Introduction and figure 2 was not consistent. Please identify and choose the better one.
  4. I am not sure that unnecessary pain is the best term.
  5. Please provide the following abbreviations: CCI (line 146), LPS (line 191), CMV (line 230), BAC (line 238), EGFP (line 239), CREB (line 291) and ROCK2 (line 318).

Author Response

Response to Reviewer 2

Comments and Suggestions for Authors

This review article addresses the role of microglial P2 receptors in pain signaling. This article discussed the importance of purinergic signaling. It also reviewed some novel purinergic agents that might control chronic pain. In my opinion, this manuscript is rigorous and provides a useful contribution to its area of research. I only have some minor queries that require additional attention. 

We are grateful to Reviewer 2 for reviewing our manuscript and giving worthful comment. Below are our responses to each comment. We believe improvement of our manuscript by your kind suggestions.

Title: I considered that the role of microglial P2 purinergic receptors in pain signaling could be better. *

We appreciate the reviewer’s suggestion. Because we described both P1 and P2 purinergic receptors, we further modified the title as follows, “The role of microglial purinergic receptors in pain signaling”.

Introduction: Please add brief sentences to describe the classification of the P1 and P2 receptors. Additionally, the relationship between P1 (not just P2) purinergic receptors and pain sensation is also required in this section. *

We appreciate and agree with the reviewer’s point. We have added 2nd sentence in 1.Introduction section, which explains the purinergic receptor family.

The abbreviation of RVM in Introduction and figure 2 was not consistent. Please identify and choose the better one. *

We appreciate the reviewer’s suggestion. We have modified figure to write “Rostral ventromedial medulla (RVM)”.

I am not sure that unnecessary pain is the best term.*

We appreciate and agree with the reviewer’s point. We have changed the word “pathological pain”.

Please provide the following abbreviations: CCI (line 146), LPS (line 191), CMV (line 230), BAC (line 238), EGFP (line 239), CREB (line 291) and ROCK2 (line 318). *

We appreciate and agree with the reviewer’s point. We have added the full names of abbreviations.

Reviewer 3 Report

In this review Tozaki-Saitoh H. et al, give an overview of the involvement of the main purinergic receptors in the regulation of pain perception, with a focus on compounds able to bind these receptors as potential pain modulators. The manuscript is quite well organized, but in my opinion some improvements are necessary before publication according to the following suggestions:

-figures with the chemical structures of the discussed compounds are necessary and must be added

-references should be updated with more recent compounds. Different new purinergic receptors modulators were published in the very last years, whose discussion should be added in the relevant sections

- The paragraph lines 347-362 at p. 8 (Conclusion section) should be moved in the introduction section, along with Figure 2, since it is a description of the purinergic receptor system which could be more useful to read at the beginning of the paper, and not at the end.

Author Response

Response to Reviewer 3

Comments and Suggestions for Authors

In this review Tozaki-Saitoh H. et al, give an overview of the involvement of the main purinergic receptors in the regulation of pain perception, with a focus on compounds able to bind these receptors as potential pain modulators. The manuscript is quite well organized, but in my opinion some improvements are necessary before publication according to the following suggestions:

We are grateful to Reviewer 3 for reviewing our manuscript and giving worthful comment. Below are our responses to each comment. We believe improvement of our manuscript by your kind suggestions.

-figures with the chemical structures of the discussed compounds are necessary and must be added*

We appreciate the reviewer’s suggestion. We have made three figures depicted the structures of compounds listed in the main text and incorporated in the revised manuscript.

-references should be updated with more recent compounds. Different new purinergic receptors modulators were published in the very last years, whose discussion should be added in the relevant sections*

We appreciate the reviewer’s suggestion. We have added explanations of new purinergic modulators in the last of each section.

- The paragraph lines 347-362 at p. 8 (Conclusion section) should be moved in the introduction section, along with Figure 2, since it is a description of the purinergic receptor system which could be more useful to read at the beginning of the paper, and not at the end.

We appreciate the reviewer’s suggestion. We have moved the figure to the first figure in the revised manuscript and added a brief explanation of the nociceptive pathway and its relation to the purinergic receptors. The most of discussion in the previous manuscript was unchanged for the purpose of recalling the scheme of the main text.

Round 2

Reviewer 3 Report

Authors have substantially improved the submitted manuscript and in my opinion it can be now accepted for publication in its present form

Author Response

Dear Reviewer 3,

We are grateful for your kind suggestions that made substantial improvements to our manuscript. 

I have corrected a mistake. The chemical structure of PSB-15417 is not disclosed. I removed the structure and mentioned it in the legend in the revised manuscript.